# Assessing Carbon Emissions from Biomass Burning in Croplands in Burkina Faso, West Africa

Pawend-taoré Christian Bougma [1,*], Loyapin Bondé [1], Valaire Séraphin Ouehoudja Yaro [1], Amanuel Woldeselassie Gebremichael [2,3] and Oumarou Ouédraogo [1]

[1] Laboratory of Plant Biology and Ecology, University Joseph Ki-Zerbo, Ouagadougou 03 B.P. 7021, Burkina Faso; loyapin.bonde@ujkz.bf (L.B.); valaire.yaro@ujkz.bf (V.S.O.Y.); ouedraogooumar@ujkz.bf (O.O.)

[2] Biodiversity Research/Systematic Botany, University of Potsdam, Maulbeerallee 1, 14469 Potsdam, Germany; amanuel.gebremichael@uni-potsdam.de

[3] Leibniz Institute for Agricultural Engineering and Bioeconomy (ATB), Max-Eyth-Allee 100, 14469 Potsdam, Germany

[*] Correspondence: christian.bougma@ujkz.bf

**Abstract:** Agricultural biomass burning plays a critical role in carbon emissions, with implications for climate change. This study aims to assess carbon (C) emissions and establish C, CO, $CO_2$ and $CH_4$ emission factors (EFs) by simultaneously testing the effects of climatic conditions and cropland category on gas emissions. In Burkina Faso, 96 experimental fires were conducted in accordance with farmers' operations during the land-clearing season in two climatic zones (Sudanian and Sudano-Sahelian) and across two cropland categories (Cropland Remaining Cropland (CC) and Land Converted to Cropland (LC)). The carbon mass balance technique was applied to estimate emissions. Climate zone and cropland category significantly influenced carbon emissions and emission factors ($p < 0.05$). The Sudanian zone recorded the highest carbon emissions ($0.24 \pm 0.01$ t C ha$^{-1}$). For cropland category, LC recorded the highest carbon emissions with an average value of $0.27 \pm 0.01$ t C ha$^{-1}$. $CO_2$ EFs ranged from $1661.44 \pm 3.63$ g kg$^{-1}$ in the Sudanian zone to $1716.51 \pm 3.24$ g kg$^{-1}$ in the Sudano-Sahelian zone. EFs showed a dependence on the cropland category, with the highest EFs in CC. Smart agricultural practices limiting cropland expansion and biomass burning need to be promoted. This study provides vital information useful for supporting decision making as part of Nationally Determined Contributions.

**Keywords:** biomass burning; carbon emission; emission factors; climatic zones; cropland category

## 1. Introduction

Biomass burning (BB) is a traditional method of land clearing widely used by farmers to prepare cultivated fields for subsequent crops [1,2]. Slashing and burning of biomass in open fields comprise a common agricultural practice used by smallholder farmers to clear cropland and convert land use [3]. Worldwide, it has become more frequent and widespread, with a significant proportion occurring in tropical Africa [4]. In West African regions where resources are limited, smallholder farming systems consisting of manual clearing and burning of biomass are mostly adopted by smallholders [5,6]. Despite the efforts to reduce agricultural small fires in most African countries [7], they are still observed in fields and have a negative impact on agricultural landscapes by increasing greenhouse gas (GHG) emissions. The use of fire by smallholder farmers is the subject of few surveys investigating its contribution to GHG emissions, and there are only a few prescribed burning programs in place.

In the context of global climate change, fire is one of the largest potential anthropogenic sources of GHG emissions [8]. Savanna landscapes represent more than half of global fire carbon (C) emissions [9]. Burning in savannas accounts for 25% of the total carbon emissions in agricultural landscapes in Africa [10], with agricultural residue burning being the

most intensive activity [11]. Fires are the cause of up to 40% of annual carbon dioxide ($CO_2$) emissions in West Africa's savannas. According to a recent report from Burkina Faso, GHG emissions associated with fires are significant at the national level, even though it may seem insignificant compared to global estimates [12]. The estimated GHG emissions resulting from crop residue burning were 3.35 kilotons for methane ($CH_4$), 0.09 kilotons for nitrous oxide ($N_2O$) and 116.72 kilotons for carbon dioxide ($CO_2$) [12]. Most GHG emission assessments are based on default emission factors (EFs) of the Intergovernmental Panel on Climate Change (IPCC) due to the lack of fine-scale EFs. This probably leads to over-estimation or underestimation of GHG emissions, which can cause errors in simulations and mislead policy making. EFs from fire are defined as the grams of trace gas emitted per kilogram of dry matter consumed during a fire [13,14]. In order to reduce uncertainty caused by the use of default EFs in GHG estimation, more and more investigations are being carried out in African savanna regions to establish specific emission factors [3,4,15]. However, little attention has been paid to the burning of agricultural biomass, despite the fact that they are the most active fires in this region [16]. Carbon dioxide ($CO_2$) is the most important anthropogenic GHG that contributes to global warming [17] and the gas that is most emitted during biomass burning. Reducing its concentrations in the atmosphere and improving mitigation actions were identified as two of the most pressing modern-day environmental issues [18]. The implementation of this recommendation requires one to have specific small-scale data and appropriate measurement methods. Mass balance techniques implementing a bottom–up approach represent one of the various GHG measurement strategies of IPCC Tier 2 used to quantify small fires [19,20]. This technique provides a measure of uncertainty associated with emissions, by calculating carbon balances [21,22]. For establishing fine-resolution EFs reflecting national circumstances, Article 13 of The Paris Agreement recommended using Tier 2 or 3 methods, especially country-specific emission factors and activity data [23]. Moreover, The Paris Agreement suggests providing information on actions, policies and measures that support the implementation and achievement of its Nationally Determined Contributions (NDCs) under Article 4 focusing on reductions in GHG emissions at the national level [24]. The development of appropriate farming practices for reducing GHG emissions in the agricultural sector is hindered by a lack of nation-specific emission data [4]. The objectives of this study are to: (i) assess the quantity of carbon emitted from open biomass burning during cropland clearing for cropping and (ii) establish emission factors for carbon as well as those for its related gases ($CO_2$, CO and $CH_4$). The main research question addressed is how climate conditions and cropland category affect carbon emissions resulting from agricultural biomass burning. To understand the effects of agricultural practices on GHG emissions, we designed our experiments according to farmers' farm field preparation. The findings of this research highlight the specific contribution of agricultural biomass burning to greenhouse gas emissions and can be used to improve both NDC and climate-smart agriculture implementations in Burkina Faso.

## 2. Methods

### 2.1. Study Area

Open agricultural biomass-burning experiments were conducted in two climatic zones of Burkina Faso: the Sudanian and Sudano-Sahelian zones. Four experimental sites were selected with two sites per climatic zone located between $9°50'$ and $13°70'$ N latitude and $00°10'$ and $5°80'$ W longitude (Figure 1). The precipitation pattern in both climatic zones is unimodal, with a prolonged dry season from November to April, followed by a short-yet-intense rainy season from May to October. According to climate data from the National Meteorology Agency of Burkina Faso, the Sudanian zone recorded a mean annual rainfall of $1153.48 \pm 203$ mm and a mean temperature of $28.15 \pm 0.25$ °C during the last decade (from 2013 to 2022). In the Sudano-Sahelian zone, mean annual rainfall and temperature values recorded during the same period were $900.40 \pm 67$ mm and $29.41 \pm 0.22$ °C, respectively. Subsistence agricultural activities are the main occupation in the study area and account for approximately 56.2% of the total population [25]. Cropland

landscapes are characterized by cropland remaining cropland (old croplands) and land converted to cropland (young croplands) with major agroforestry species such as *Vitellaria paradoxa* C.F.Gaertn., *Parkia biglobosa* (Jacq.) Benth. and *Lannea macrocarpa* Engl. and K. Krause for the Sudanian zone and *Vitellaria paradoxa, Parkia biglobosa, Lannea macrocarpa, Sclerocarya birrea* (A. Rich.) Hochst. and *Tamarindus indica* L. For the Sudano-Sahelian zone [26]. Due to the growing demand for arable land, the expansion of agriculture through forest clearance is the main driver of deforestation in the study area [27]. Annual field clearing using biomass burning is generally carried out at the end of the dry season between April and May. Field clearing includes traditional slash-and-burn methods to clear plant biomass (shrubs, plant regeneration and crop residues).

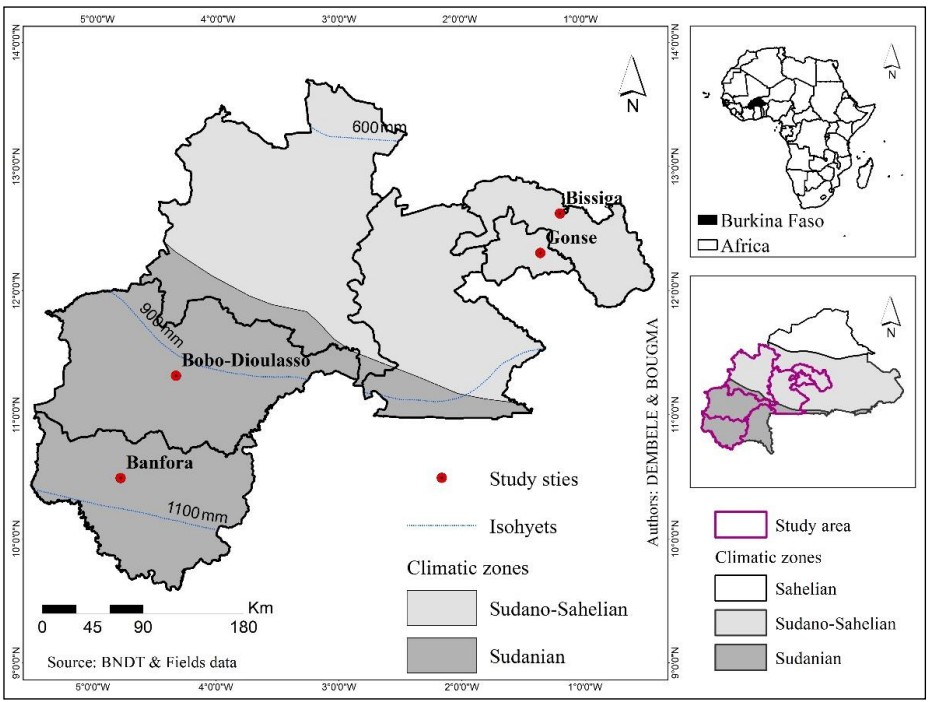

**Figure 1.** Location of the study area showing the main study sites.

### 2.2. Methodological Approach

Prior to establishing the experimental sites, a survey was conducted in the study regions to observe farmers' land management practices, which led to the identification of representative sites relevant for biomass assessment and real-time burning experiments. Two cropland categories were selected based on their spatial distribution and their accessibility: Land Converted to Cropland (LC) and Cropland Remaining Cropland (CC). Experimental plots were established in each cropland category and data were collected in two steps: biomass assessment in croplands and carbon determination in pre- and post-fire biomasses.

### 2.3. Sampling Design for Biomass Assessment

Field sampling was conducted on-farm between April and May during the time when most farmers were preparing their fields. A stratified sampling design based on climatic conditions and cropland age was adopted for the selection of sampling plots. These factors were found to have a significant influence on fire behavior [28,29]. Cropland classification was based on IPCC criteria [30]: Lands Converted to Croplands are croplands less than 20 years old and Croplands Remaining Croplands are croplands more than 20 years old. Within each climatic zone, two sampling sites were selected to account for site variability. In total, 92 main plots of dimensions 100 m × 100 m (10,000 m$^2$) were set up across the two climatic zones with 9 to 14 plots per cropland category for biomass assessment (Figure 2).

To account for biomass variability within each plot, 6 subplots of 25 m$^2$ (5 m × 5 m) at least 10 m apart were set up with 3 subplots located under tree canopies and 3 located outside tree canopies (Figure S1 in Supplementary Materials). Farmers' biomass burning during cropland clearing includes the burning of crop residues, plant regeneration and foliage from defoliation of agroforestry trees preserved in croplands. The main part of foliage biomass is usually located under the tree canopy; hence, it is important to consider this factor in biomass assessment. At the site level, a minimum distance of 1 km was observed among main plots to account for variation in soil types, crop varieties and plant species. The total number of adult trees was counted and their morphological traits such as total height, stem diameter at breast height (DBH) and small and large crowns were measured. This information was used to determine tree density and tree canopy cover per plot.

| Climatic zones | Sudanian | | | | Sudano-Sahelian | | | |
|---|---|---|---|---|---|---|---|---|
| Sites | Bobo-Dioulasso | | Banfora | | Gonsé | | Bissiga | |
| Cropland categories | LC | CC | LC | CC | LC | CC | LC | CC |
| Plots numbers | 11 | 12 | 13 | 9 | 9 | 14 | 10 | 14 |

**Figure 2.** Experimental design showing plot distribution according to climatic zones and cropland categories.

*2.4. Fuel Biomass Assessment*

During the clearing season, existing fuel biomass on fields is harvested and sun dried in situ for at least one to two weeks before burning to ensure that most of the biomass is potentially burnable (Figure 3). We intended to simulate the same conditions as farmers for biomass burning by adopting the method used by [20,31]. Most plant regeneration found in each subplot was manually slashed with machetes. Dry fuel included dead wood, crop residues, dead leaves and grass. Total dry biomass (pre-fire biomass) in each subplot was separately weighed using an electronic balance (weight = 0–5 kg, with precision = 1 g) according to fuel size, i.e., fine fuel (components with diameters < 0.60 cm) and large fuel (diameters components > 0.60 cm) [21,32]. Dead wood (diameter > 2.5 cm) was removed in our case because it is generally harvested by farmers and used as domestic firewood [5]. Diameters of fuel components were measured using vernier calipers.

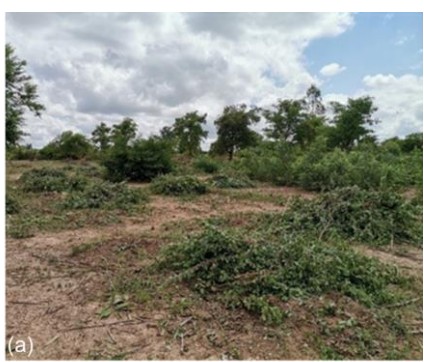 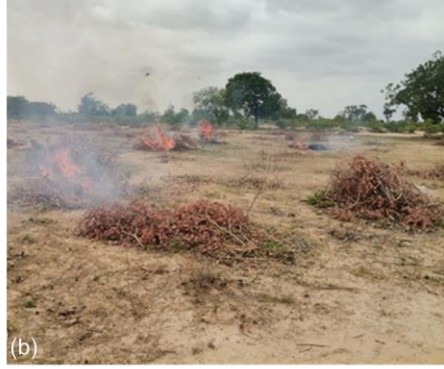

**Figure 3.** Cropland clearing according to farmers' practices: (**a**) biomass slashed and piled in cropland converted to cropland, (**b**) fire ignition for biomass burning.

*2.5. Burning Experimentation*

After biomass weighing, experimental fires were set on the investigated fields. To account for farmers' practices, all biomass components from the subplots (under and outside the canopy) were mixed, and two composite biomass samples of 2 kg were taken to be

burned on a steel tray of 4 m² (2 m × 2 m). Composite samples were burned separately for the determination of the carbon content. Fires were ignited in the direction of the wind as practiced by farmers between 4 and 5 pm using a bunch of dried grass biomass to ensure rapid linear ignition. Once the residues left behind had cooled sufficiently after each experimental fire, all biomass post-fire (ash, charcoal, unburned fuels) remains on the steel tray were collected and weighed to determine the amount of biomass consumed. The experimental procedure is illustrated in Figure 4. In total, 96 experimental fires were carried out according to the sampling design: 2 climatic zones × 2 sites × 2 cropland categories × 6 plots × 2 fire experiments, resulting in the collection of 96 composite samples pre-fire and post-fire. Pre-fire composite samples consisted of 0.1 kg of biomass randomly collected from 36 plots while post-fire composite samples consisted of 0.05 kg of biomass randomly collected from 36 experiments. Possible unburned and charcoal fuels were separated according to the type of fuel components as mentioned above to estimate the fuel loss of each component. All composite samples collected were brought to the laboratory for moisture content and carbon content determination.

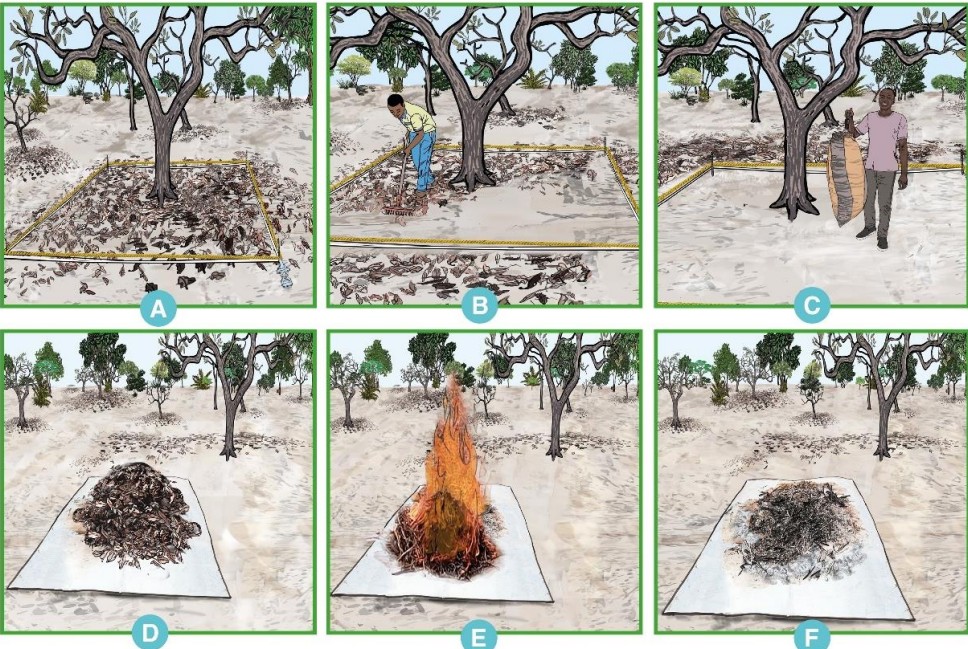

**Figure 4.** An experimental procedure illustrates biomass burning in field. (**A**,**B**) biomass collection, (**C**) weighing under the tree canopy, (**D**,**E**) experimental fire ignition and (**F**) post-fire sample (ash, charcoal, unburnt fuel).

### 2.6. Carbon Content Determination

Carbon contents in plant materials were determined using the ash method [33]. Pre- and post-fire composite samples were oven-dried at 65 °C until a constant weight was obtained for carbon content determination. The standard calculation of moisture content values was used. Duplicate samples of 2 g of dry powder of each component were incinerated at 550 °C for 2 h. After calcination and cooling, the carbon contents were assessed according to Equations (1) and (2).

$$\text{Ash}(\%) = \frac{W_3 - W_1}{W_2 - W_1} \times 100 \tag{1}$$

$$\text{Carbon }(\%) = (100\% - \text{Ash}(\%)) \times 0.58 \tag{2}$$

where $w_1$ is the crucible weight, $w_2$ is the weight of crucibles with samples, $w_3$ is the weight of crucibles and ash and 0.58 is the content of carbon in dry organic matter.

### 2.7. Data Analysis

2.7.1. Tree Structural Parameters

Trees' ecological parameters such as tree density, tree canopy cover and biomass under canopy were calculated. At site and climatic zone levels, tree density per hectare was calculated by computing the mean value of the number of trees recorded in the individual plots considered.

Individual tree canopy cover ($S_i$) was defined as the proportion of soil covered by the vertical projection of leafy crowns of individual trees. $S_i$ was estimated using Equation (3), which is widely used for the determination of the vegetation cover of most species in agroforestry parks in West Africa [34]. Tree canopy cover at the main plot level (S) was calculated by summing individual $S_i$ within the main plot. In each plot, the mean biomass under canopy ($B_s$) was estimated in the three subplots and reported per $m^2$. The total biomass under the tree canopy per main plot was then estimated using Equation (4).

$$S_{i\ canopy} = \frac{\Pi(d_1 \times d_2)}{4} \tag{3}$$

With $S_i$: canopy area ($m^2$) at individual tree level; $d_1$: small crown (m); $d_2$: large crown (m).

$$B_{UC} = B_s \times S \tag{4}$$

With $B_{UC}$: biomass under canopy, $B_S$: mean biomass from three subplots of 25 $m^2$, S: the summation of individual trees' canopy cover (*Si canopy*) in the main plot ($m^2$).

2.7.2. Carbon Content and Fuel Characteristics Calculations

Assessment of carbon content in fuel biomass was conducted both pre- and post-fire by considering: (i) the amount of biomass fuel before burning ($B_{pre}$) and its corresponding carbon content ($CC_{pre}$ (%)) and (ii) biomass after burning ($B_{post}$) with its carbon content ($CC_{post}$ (%)); both were determined in the laboratory. The pre-fire carbon load ($C_{pre}$, t ha$^{-1}$) and the post-fire carbon load ($C_{post}$, t ha$^{-1}$) were calculated using Equation (5) and Equation (6) [21], respectively.

$$C_{pre} = B_{pre} \times CC_{pre} \tag{5}$$

$$C_{post} = B_{unburnt} \times CC_{unburnt} + B_{charcoal} \times CC_{charcoal} + B_{ash} \times CC_{ash} \tag{6}$$

Post-fire biomass included unburnt biomass ($B_{unburnt}$), charcoal biomass ($B_{charcoal}$) and ash biomass ($B_{ash}$). $CC_{unburnt}$: carbon content in unburnt fuel; $CC_{charcoal}$: carbon content in charcoal; and $CC_{ash}$: carbon content in ash.

2.7.3. Fuel Characteristics Calculations

Calculation of Percentage of Biomass and Carbon Loss

The percentage of carbon loss during the burning was determined by comparing the mass of carbon before and after fire exposure according to Equation (7) [35].

$$Carbon loss(\%) = 100 \times (1 - \frac{C_{post}}{C_{pre}}) \tag{7}$$

Calculation of Percentage of Carbon Remaining in Post-Fire Sample

The percentage of the C remaining ($C_{rem}$) in samples post-fire compared to pre-fire C was calculated using Equation (8) [35]. Crem also represents the conversion rate of pre-fire carbon to post-fire carbon.

$$C_{rem}(\%) = 100 \times \frac{C_{post}}{C_{pre}} \tag{8}$$

Calculation of Combustion Completeness

Combustion completeness is an important factor used in the majority of fire studies to estimate biomass burning emissions, with information on data activity. The combustion completeness (CC) (%), commonly called the combustion factor, is the fraction of the total biomass exposed to fire that actually burned [28]. It was calculated using Equation (9).

$$CC(\%) = \frac{(C_{ash} + C_{charcoal} + C_{emitted})}{C_{pre}} \tag{9}$$

With CC (%): combustion completeness, $C_{ash}$: ash carbon load, $Cc_{charcoal}$: charcoal carbon load, $C_{emited}$: carbon emitted during fire exposure.

### 2.7.3.4. Carbon Budget Calculations
Calculation of Carbon Emissions

To assess carbon emissions and carbon-related gases, we used the burnt carbon approach described above. This approach is based on changes between the pre- and post-fire carbon contents of the fuels [36,37]. Carbon emitted ($C_{emit}$) into the atmosphere was then estimated as the difference between the pre-fire fuel load and post-fire fuel load [21]. Carbon emissions ($C_{emission}$) were calculated using Equation (10) [37].

$$C_{emission} = \frac{(C_{pre} - C_{post})}{C_{pre}} \times (C_{pre} - C_{post}) \tag{10}$$

$C_{pre}$ and $C_{post}$ represent pre-fire and post-fire carbon; $\Sigma C_{emit}/C_{fuel}$ represents the correction factor required for the accounting framework of burnt carbon.

Estimation of Carbon Dioxide Equivalent from Carbon Emitted

Emission of $CO_2$ equivalent ($CO_2\_eq$) based on conversion from the carbon emitted was calculated using Equation (11) [34].

$$CO_2\_eq = \text{Total carbon} \times 3.67 \tag{11}$$

where total carbon is the mass of the carbon emitted and 3.67 is the ratio of the molecular masses of $CO_2$ and C (44/12). Total carbon was converted to $CO_2$, assuming that the other carbon-related gases (CO, $CH_4$ and non-methane hydrocarbons (NMHC) were also subsequently oxidized to $CO_2$ [38].

### 2.7.3.7. Determination of Emission Factors

The basic definition of a fuel-based emission factor (EF) is the mass of compound released per unit mass of fuel consumed [39]. The carbon EF expressed in grams of carbon emitted per kilogram of dry matter burned was computed using Equation (12) [32].

$$EF = \frac{\sum C_{emit}}{C_{fuel}} \tag{12}$$

where $\sum C_{emit}$ is the mass of fuel carbon emitted into the atmosphere and represents the sum of carbon contained in the emitted carbon gas species ($CO_2$, CO, $CH_4$, NMHC and PC) and $C_{fuel}$ represents the mass of fuel burnt. Based on the approach used by [40], we calculated the percentage of carbon lost corresponding to the carbon emission factor. Then, we applied this percentage to determine the mass of carbon emitted per unit of dry fuel consumed. The percentage of carbon loss was multiplied by the conventional conversion factor ($1000 \text{ g kg}^{-1}$) to determine the mass of carbon emitted per kg of dry fuel consumed.

EFs for $CO_2$, CO and $CH_4$ were calculated using the carbon mass balance method [28,37]. As indicated above, the mass of carbon burned was emitted as carbon gas species such as $CO_2$, CO, $CH_4$ and NMHC. The mass of carbon can be approximated as the sum of three

main gases: $CO_2$, CO and $CH_4$ because they typically account for 97–99% of total carbon emissions [13,14,41].

Using the emission factors for $CO_2$, CO and $CH_4$ synthesized by [13], we converted these EFs (given in gas mass) into carbon EFs and then calculated the fraction of carbon emitted as $CO_2$, CO or $CH_4$, included in the mass balance. The mass of carbon contained in each gas was calculated using the molar number and molar mass of each gas. The percentages of $CO_2$, CO and $CH_4$ content in total carbon after different calculations were estimated to be 91.37%, 7.63% and approximately 1%, respectively. The calculation steps are summarized in Table 1.

**Table 1.** Emission factors from [13] summarizing emission factors of many studies investigating burning of agricultural residues.

| | $CO_2$ | CO | $CH_4$ | $NMOG_S$ | OC | EC/BC | Total |
|---|---|---|---|---|---|---|---|
| Emission factors (g kg$^{-1}$) | 1430 | 76 | 5.7 | 51 | 4.9 | 0.42 | 1568.02 |
| Number of studies (*n*) | 29 | 39 | 20 | 0 | 20 | 24 | |
| % of species mass | 91.20 | 4.85 | 0.36 | 3.25 | 0.31 | 0.03 | 100 |
| Molar mass of species | 44 | 28 | 16 | Undefined | Undefined | Undefined | - |
| Number of mole species | 32.50 | 2.71 | 0.36 | - | - | - | - |
| Mass of carbon (g) | 390 | 32.57 | 4.28 | - | - | - | 426.85 |
| % carbon of species | 91.37 | 7.63 | 1.00 | - | - | - | 100 |

We calculated EFs of $CO_2$, CO and $CH_4$ using Equation (13) as mentioned by [3,13].

$$EF_i = F_c \times \frac{MM_i}{AM_c} \times \frac{C_i}{C_{total}} \times 1000 \text{ g kg}^{-1} \qquad (13)$$

where $EF_i$ is the mass of gas species i emitted per kg of dry fuel consumed (g/kg), $F_C$ is the fractional fuel carbon content (the majority of the fuel carbon content in this study varied between 0.52 to 0.55; we used the carbon content value corresponding to each factor: climate zone and cropland category), 1000 is a unit conversion factor (1000 g kg$^{-1}$), $MM_i$ is the molecular mass of gas species i, $AM_c$ is the atomic mass of carbon and $C_i/C_t$ is the number of moles of gas species i emitted divided by the total number of moles of carbon emitted.

2.7.3.8. Statistical Analyses

For data analysis, carbon content (%), fuel biomass (t ha$^{-1}$), moisture content (%), carbon loss (%), combustion completeness (%), carbon remaining post-fire (%), carbon emissions (t ha$^{-1}$) and gas emission factors (g kg$^{-1}$) were considered as response variables while climate zone and cropland category were considered as explanatory variables. General Linear Models (GLM) were used to test the effects of explanatory variables on response variables. Specifically, GLM with binomial error (logit link function) were applied for response variables reported as percentages (%). GLM with gamma error (identity link function) were applied for response variables reported as masses. When a significant effect was detected, the Wilcoxon test was used to examine the effect of cropland age on response variables within climate zones. For all tests, statistical significance was set at 5%. All statistical analyses were performed with R software (Version R4.2.3) [42].

**3. Results**

*3.1. Effect of Climatic Zone and Cropland Category on Fuel Biomass, Carbon and $CO_2$ Emissions*

GLM showed that climatic zone and cropland category have separate significant effects on fuel biomass, carbon and $CO_2$ emissions (Table 2). Regarding climatic zone, the highest values were observed in the Sudanian zone with a mean biomass estimated at $0.50 \pm 0.02$ t ha$^{-1}$ corresponding to $0.24 \pm 0.01$ t C ha$^{-1}$ and $0.89 \pm 0.05$ $CO_2$ eq ha$^{-1}$ emitted. The Sudano-Sahelian zone recorded the lowest values with a mean biomass estimated at $0.34 \pm 0.02$ t ha$^{-1}$ corresponding to $0.17 \pm 0.01$ t C ha$^{-1}$ and $0.63 \pm 0.04$ t $CO_2$ eq ha$^{-1}$

emitted (Table 3). Fuel biomass, carbon and $CO_2$ emissions showed significant differences between cropland categories, regardless of the climatic zone. The highest values of biomass, carbon and $CO_2$ eq emitted were observed in LC (Table 3). Results from the Wilcoxon test within each climate zone indicated that the cropland category had a significant effect on all these variables (Figure 5). Detailed information regarding variability in fuel characteristic traits such as carbon content in different fuel components, moisture content, carbon loss, combustion completeness and carbon remaining in post-fire fuel components are provided in Table S1 in Supplementary Materials.

**Table 2.** GLM testing the effect of climatic zones and cropland categories on biomass, C and $CO_2$.

| Predictors | Estimates | Standard Error | t-Value | Pr(>|t|) |
|---|---|---|---|---|
| **Biomass** | | | | |
| Intercept | 0.658 | 0.040 | 16.477 | <0.0001 |
| Climatic zones | −0.160 | 0.032 | −4.886 | <0.0001 |
| Cropland categories | −0.280 | 0.038 | −7.248 | <0.0001 |
| **Carbon emission** | | | | |
| Intercept | 0.314 | 0.019 | 16.073 | <0.0001 |
| Climatic zones | −0.070 | 0.016 | −4.225 | <0.0001 |
| Cropland categories | −0.131 | 0.020 | −6.829 | <0.0001 |
| **$CO_2$ eq emission** | | | | |
| Intercept | 1.153 | 0.071 | 16.075 | <0.0001 |
| Climatic zones | −0.253 | 0.060 | −4.227 | <0.0001 |
| Cropland categories | −0.481 | 0.070 | −6.829 | <0.0001 |

**Table 3.** Effect of climatic zones and cropland categories on fuel biomass, carbon and $CO_2$ emissions (mean ± SE).

| Characteristics | Climatic Zones | | Cropland Categories | |
|---|---|---|---|---|
| | Sudanian | Sudano-Sahelian | LC | CC |
| | *n* = 45 | *n* = 47 | *n* = 43 | *n* = 49 |
| Biomass (t ha$^{-1}$) | 0.50 ± 0.02 [a] | 0.34 ± 0.02 [b] | 0.57 ± 0.03 [a] | 0.29 ± 0.01 [b] |
| Carbon emission (t ha$^{-1}$) | 0.24 ± 0.01 [a] | 0.17 ± 0.01 [b] | 0.27 ± 0.01 [a] | 0.14 ± 0.01 [b] |
| $CO_2$ eq emitted (t ha$^{-1}$) | 0.89 ± 0.05 [a] | 0.63 ± 0.04 [b] | 1.02 ± 0.06 [a] | 0.53 ± 0.03 [b] |

*Values with different letters indicate significant differences (p < 0.05, Wilcoxon test) between climatic zone and cropland category; n = number of plots, SE = Standard Error, CC: Cropland Remaining Cropland, LC: Land Converted to Cropland.*

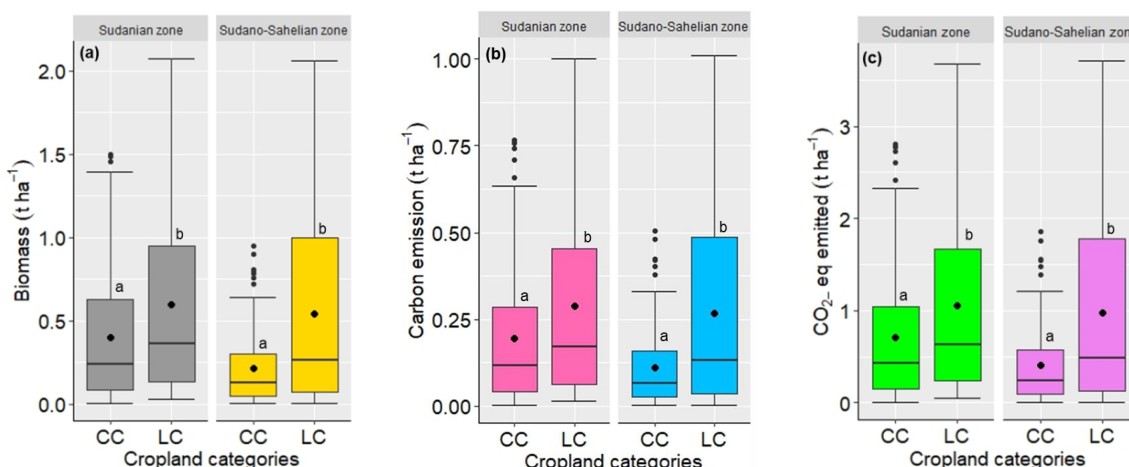

**Figure 5.** Variability in biomass, carbon emitted and carbon dioxide emitted across cropland category (CC: Cropland Remaining Cropland; LC: Land Converted to Cropland) per climatic zone in Burkina Faso. The median and mean are represented by a horizontal line and a dot in the box plots, respectively. Different letters above the boxes indicate significant differences between cropland categories for each climatic zone (*p* < 0.05).

### 3.2. Effect of Climatic Zone and Cropland Category on Emission Factors of C, $CO_2$, CO and $CH_4$

The emission factors of C, $CO_2$, CO and $CH_4$ were significantly influenced by climatic zones and cropland category ($p < 0.0001$, Table 4). The highest values were recorded in the Sudano-Sahelian zone and the CC category (Table 5). Similarly, the EFs of other carbon components were highest in the Sudano-Sahelian zone in the CC category, with C, $CO_2$, CO and $CH_4$ EFs of $963.38 \pm 0.34$ g kg$^{-1}$, $1757.09 \pm 1.75\pm$ g kg$^{-1}$, $146.38 \pm 0.14 \pm$ g kg$^{-1}$ and $19.16 \pm 0.01$ g kg$^{-1}$, respectively (Figure 6 a–d).

**Table 4.** GLM presenting effects of climatic zones and cropland categories on C, $CO_2$, CO and $CH_4$ EFs.

| Predictors | Estimates | Standard Error | t-Value | Pr(>|t|) |
|---|---|---|---|---|
| **Carbon emission factor** | | | | |
| Intercept | 894.574 | 0.60 | 1492.15 | <0.0001 |
| Climatic zones | 8.180 | 0.725 | 11.27 | <0.0001 |
| Cropland categories | 60.606 | 0.725 | 83.5 | <0.0001 |
| **$CO_2$ emission factor** | | | | |
| Intercept | 1613.47 | 1.844 | 875.12 | <0.0001 |
| Climatic zones | 41.964 | 2.242 | 18.72 | <0.0001 |
| Cropland categories | 102.633 | 2.242 | 45.78 | <0.0001 |
| **CO emission factor** | | | | |
| Intercept | 134.42 | 0.153 | 875.12 | <0.0001 |
| Climatic zones | 3.496 | 0.186 | 18.72 | <0.0001 |
| Cropland categories | 8.550 | 0.186 | 45.78 | <0.0001 |
| **$CH_4$ emission factor** | | | | |
| Intercept | 17.598 | 0.020 | 875.12 | <0.0001 |
| Climatic zones | 0.457 | 0.024 | 18.72 | <0.0001 |
| Cropland categories | 1.119 | 0.024 | 45.78 | <0.0001 |

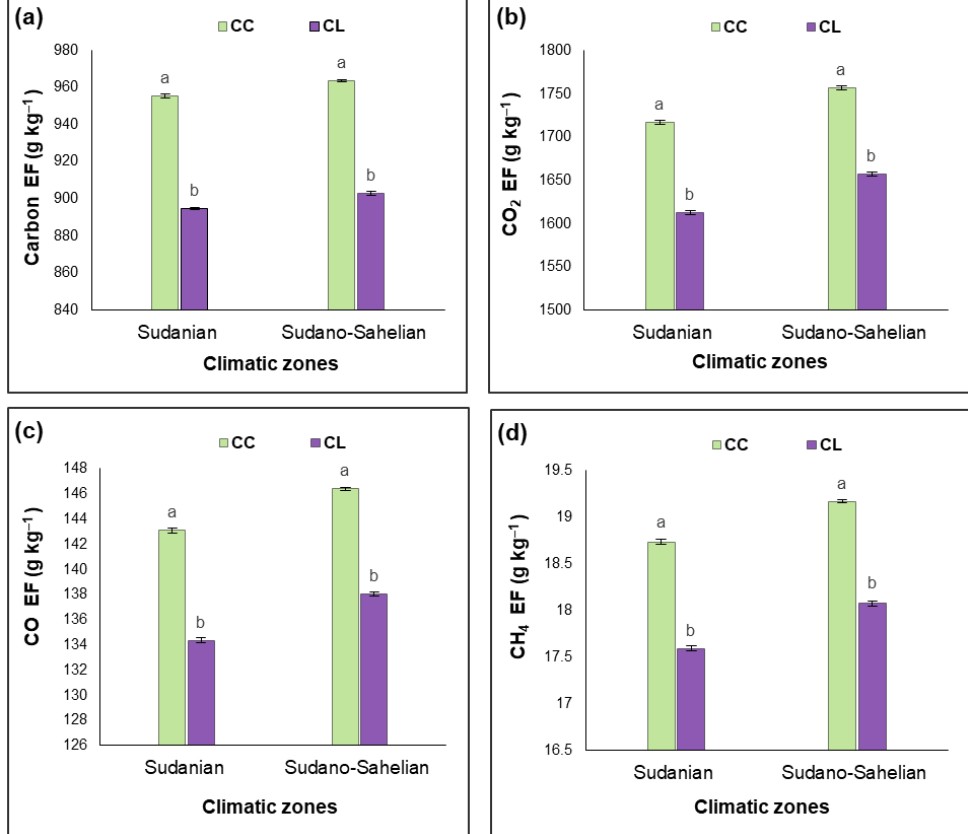

**Figure 6.** Emission factors (g kg$^{-1}$) of C (**a**), $CO_2$ (**b**), CO (**c**) and $CH_4$ (**d**) (mean $\pm$ SE) of LC and CC in the two climatic zones during the fire experimentation. Different letters above the bars indicate significant differences between cropland categories for each climatic zone ($p < 0.05$). Error bars show standard errors.

**Table 5.** Effect of climatic zones and cropland categories on C, $CO_2$, CO and $CH_4$ emission factors (mean $\pm$ SE).

| Fuel Characteristics | Climatic Zones | | Cropland Categories | |
|---|---|---|---|---|
| | Sudanian | Sudano-Sahelian | LC | CC |
| | *n* = 45 | *n* = 47 | *n* = 43 | *n* = 49 |
| Carbon emission factor (g kg$^{-1}$) | 922.85 $\pm$ 1.93 [a] | 938.86 $\pm$ 1.82 [b] | 898.18 $\pm$ 0.57 [a] | 959.85 $\pm$ 0.55 [b] |
| $CO_2$ emission factor (g kg$^{-1}$) | 1661.44 $\pm$ 3.63 [a] | 1716.51 $\pm$ 3.24 [b] | 1632.04$\pm$ 2.15 [a] | 1740.06$\pm$ 1.86 [b] |
| CO emission factor (g kg$^{-1}$) | 138.41 $\pm$ 0.3 [a] | 143.01 $\pm$ 0.27 [b] | 135.96$\pm$ 0.17 [a] | 144.96$\pm$ 0.15 [b] |
| $CH_4$ emission factor (g kg$^{-1}$) | 18.12 $\pm$ 0.03 [a] | 18.72 $\pm$ 0.03 [b] | 17.80 $\pm$ 0.02 [a] | 18.97$\pm$ 0.02 [b] |

*Note: The post-fire samples selected for chemical analysis were fully charred for the old fields and not fully charred for the young fields.* Values with different letters indicate significant differences ($p < 0.05$, Wilcoxon test) between climatic zone and cropland category. *n* = number of plots, SE = Standard Error.

## 4. Discussion

### 4.1. Effect of Climatic Zone on Carbon Emissions and Gas Emission Factors

Fuel biomass, carbon emissions and emission factors were significantly different between the two climatic zones with the highest values recorded in the Sudanian zone. Carbon and $CO_2$ emissions are highly and positively correlated to the amount of fuel biomass. Thus, ecological factors influencing biomass production also affect both carbon and $CO_2$ emissions. Fuel biomass from agricultural lands is mainly composed of crop residues, dry foliage, grass and juveniles from tree regeneration. The Sudanian zone has a more favorable climate (especially considering its high rainfall) for biomass production than the Sudano-Sahelian zone. The more humid climatic conditions in West Africa lead to higher grass biomass and crop yield, including crop residues [43]. Grass biomass production decreases across the climatic gradient of Burkina Faso, and there is a positive correlation between biomass and annual rainfall [44]. Humid areas are more likely to have the potential for tree regeneration and vegetation productivity than drier areas [45]. In a previous study [46], it was found that the intensity of fires in burned areas increased with rainfall, which suggests that dry biomass increased in areas with high rainfall. Dri et al. [20] showed that carbon emissions vary depending on the agro-ecological zone. Appropriate management of biomass and fire in Sudanian croplands is crucial for reducing $CO_2$ emissions emitted into the atmosphere. The carbon emissions reported in the present study ranged from 0.14 to 0.27 t C ha$^{-1}$, which is noticeably low compared to the carbon emissions of other land-use types in different landscapes [20,21]. The highest emission factor was recorded in the Sudano-Sahelian zone where strong wind speeds, high temperatures and low air relative humidity were observed compared to the Sudanian zone. Due to these climate conditions, fuel biomass water content is reduced and combustion completeness is improved, leading to high carbon and associated gas species emission factors. Wind speed is a crucial factor in determining flashover conditions in open burning under field conditions and resulted in the emission of most of the carbon in the fuel burned by the fire, with only a minor pyrogenic carbon component remaining after the fire [21]. Fires in the Sudanian zone tend to be low in intensity, because the biomass dominated by the fuel layer retains moisture from the previous wet season. The moisture content of grasses influences the EFs [15].

### 4.2. Effect of Cropland Categories on Carbon Emissions and Emission Factors

Cropland category has a significant effect on fuel biomass, carbon and $CO_2$ emissions, with the highest values observed in LC and the lowest in CC.

Given that LC are sourced from natural vegetation or fallows, they produce a significant amount of regeneration [27,47] due to the presence of living remnant trees and shrubs [48]. The fuel biomass to be burned during annual field clearing is increased by this regeneration which leads to larger carbon emissions in LC than in CC. Fuel biomass decreases gradually from year to year because of recurrent and intensive agricultural activities, so the biomass is considerably reduced after 20 years of cultivation. Previous studies in

Burkina Faso [27,49] showed that anthropogenic slash-and-burn activities [50] converting near-natural vegetation to cropland contribute to a reduction in tree density, vegetation regeneration and, subsequently, plant biomass in agricultural landscapes. When lands are cleared or disturbed to accommodate new crops and once biomasses typically composed of piled woody debris or regeneration are burned, C is displaced from plant biomasses into the atmosphere via combustion or decay [47,51]. Results from this work indicated that biomass burning in LC significantly increases GHG emissions into the atmosphere, contributing to global warming. In West Africa, the demand for arable land is increasing due to the adverse effects of climate change and rapid human population growth [52,53]. As a result, more and more natural vegetation is being cleared for the installation of new fields [27]. It is essential to develop smart agricultural practices that limit both cropland expansion and biomass burning during field preparation. For example, the application of crop residues in agricultural soils has been shown to mitigate climate change by enhancing soil carbon storage and soil fertility. In addition, it is highly recommended to increase farmer awareness about GHG emissions resulting from biomass burning as most farmers are focused on increasing crop yield while neglecting or giving little attention to GHG emissions.

Gas species EFs in agricultural landscapes are significantly influenced by cropland age, with higher values observed in CC. This difference could be due to the fact that the biomasses in 20-year croplands are mainly composed of fine fuels (components with a diameter of less than 0.60 cm) such as tree foliage, grass, litter, bark, twigs and crop residues. In contrast, the biomasses in young fields are usually composed of large-sized fuels due to vegetation regeneration and fragmentation of the surrounding natural vegetation [45]. Due to their rapid drying response to higher temperatures and lower relative humidity, fine fuels are usually completely consumed by most fires [46,54], resulting in complete combustion and high emission factors. These results are similar to those of previous studies [14,55] which indicated that fine fuels tend to have a larger value of combustion completeness than larger-diameter biomass components. They generally produce more PyC than fine fuels, increasing the amount of carbon mass remaining in the soil after the fire [54,56]. Our results are in line with these findings [16,56], indicating that not all the available biomass in the landscape is burned during fires.

Emission factors calculated in our study are different from values found in the literature and summarized by [13] for field measurement. The average emission factors of $CO_2$, CO and $CH_4$ in open burning were higher compared to Andreae's summary of EFs for agricultural residues. These differences could be explained by the difference of carbon content in pre-fire biomasses. Other studies reported carbon fractions ranging from 0.425 to 0.50% [3,57] while our values are relatively higher (between 0.52 to 0.55%). Biomass burning practices can also influence the amount of gas emitted per unit of dry biomass. In our case, biomass was obtained from field preparation, i.e., the burning of cleared biomass before the cropping, whereas most studies focused on the burning of residues produced after the harvest. Laris et al. [58] showed that leaf fall is another factor influencing emission factors. Some studies reported a high variability of CO and $CH_4$ EFs, which can be attributed to spatial variability, vegetation type and burning time [57,59]. Multiple factors such as fuel moisture (affecting combustion efficiency) as well as intrinsic physicochemical fuel properties (e.g., lignin content, fuel species, bulk density) can affect these variabilities. Our fieldwork observations showed a higher apparent density and higher lignin content in large-sized fuels from LC compared to fine fuel represented primarily in CC. Our estimation of EFs based on carbon mass balance techniques is higher than the values reported in similar field studies. [28]. This difference can partly be explained by the differences in the methods applied to measure emissions and calculate EFs. Laboratory or field inventories to estimate EFs can influence the variability of gas species EFs. Laboratory-scale experiments, however, can produce fire characteristics that are considerably different from those of natural behavior, which can lead to the overestimation or underestimation of EFs [60]. Estimations of $CO_2$ and CO EFs derived from burning experiments in Africa [15,59] and

West Africa are [3] within the range of EFs obtained in our study. Our $CO_2$ EF values are comparable to the values reported for West African savanna $CO_2$ EFs [57]. The mean EFs for CO and $CH_4$ obtained in [3,15] according to local practices of savannas in Mali, West Africa, were both low, compared to our finding. More attention should be given to small fires in agriculture. Field-based investigations are necessary to determine the contribution of fire to GHG emissions as these emissions could be greatly underestimated. Fires in agricultural fields are usually short-lived and often undetectable from space.

## 5. Conclusions and Implications

With the objective of reducing the large uncertainties in the estimation of emissions resulting from biomass burning, we presented an improved estimation of emission factors from agricultural practices. This study provided the first direct measurements of carbon emissions from small agricultural fires that are a common practice in smallholder agriculture in West Africa. Estimation of carbon emissions and EFs from field campaigns indicated significant variability associated with climatic conditions and cropland category. Based on the amount of carbon emitted per climatic zone, the Sudanian zone released the largest quantity of carbon. The highest carbon emissions were observed in croplands under 20 years old. The highest levels of carbon emissions and the lowest emission factors are attributed to ecological conditions including rainfall, wind speed, plant regeneration, soil fertility, cropland age and fuel characteristics. The emission factors of gas species were inversely related to their emissions. This study showed that the EF of $CO_2$ was higher than those reported in most studies in the savanna, which could be explained by the differences in carbon content in fuel, farmers' practices, fuel size, fuel composition and fire intensity. The burning of cleared biomass using small fires in local practices, particularly in the Sudanian zone and in land converted to cropland, is a significant contributor to GHG emissions. Data from long-term experiments are needed to be able to assess the annual variability of carbon emissions. Our field measurements were thought to be a more accurate representative of the real environmental conditions of biomass burning by farmers on croplands.

This paper provides realistic information on carbon and emission factors, which can be used to provide better estimates of emissions at a national level. We suggest the use of specific EFs from climatic zones and croplands for accurate emission estimates at the local level. Our findings could be useful in informing and supporting decision making in fire management in agricultural landscapes for reducing carbon emissions.

**Terminology:** In this study, Land Converted to Cropland (LC) refers to land with less than 20 years of cultivation after the conversion from near-natural vegetation, forest or fallow to cropland. Other terms such as new cropland, young fields, new fields and land recently converted to cropland refer to LC. Cropland Remaining Cropland (CC) refers to land with more than 20 years of cultivation. Terms old fields and old croplands refer to this type of land. The term cropland category was derived from the IPCC [30] to refer to LC and CC.

**Supplementary Materials:** The following supporting information can be downloaded at: https://www.mdpi.com/article/10.3390/fire6100402/s1, Table S1. Biophysical characteristics (mean ± SE) of fuel biomass in the fire experimental sites; Figure S1. Experimental design showing subplots (5 m × 5 m) distribution within main plot (100 m × 100 m).

**Author Contributions:** This study and its manuscript were led and conceptualized by O.O., L.B., A.W.G. and P.-t.C.B. V.S.O.Y. was involved in the fieldwork and data organization as well as discussing and commenting on the manuscript. L.B. and P.-t.C.B. analyzed the results and drafted the manuscript. O.O. and A.W.G. provided scientific and critical guidance and drafted and reviewed the final version of the manuscript. All authors have read and agreed to the published version of the manuscript.

**Funding:** This study was funded by the German Federal Ministry of Education and Research (BMBF) through the West African Science Service Centre on Climate Change and Adapted Land Use (WASCAL) WRAP 2.0 projects (reference number: 01LG2081A). We are also grateful to the Ministry of Environment, Water and Sanitation for logistic support through the Capacity Building Initiative for Transparency (CBIT).

**Institutional Review Board Statement:** Not applicable.

**Informed Consent Statement:** Not applicable.

**Data Availability Statement:** Data sets analyzed in this study are available upon reasonable request.

**Acknowledgments:** The authors are grateful to the German Federal Ministry of Education and Research (BMBF) and the West African Science Service Centre for Climate Change and Adapted Land Use (WASCAL) through WRAP 2.0 projects for funding this study. We cordially thank field guides and local populations for their assistance.

**Conflicts of Interest:** The authors declare no conflict of interest.

## Abbreviations

| | |
|---|---|
| BB | Biomass burning |
| C | Carbon |
| $CO_2$ | Carbon dioxide |
| $CO_2$eq | Carbon dioxide equivalent |
| CO | Carbon monoxide |
| $CH_4$ | Methane |
| CC | Cropland Remaining Cropland |
| EFs | Emission factors |
| GHG | Greenhouse gases |
| GLM | General Linear Models |
| IPCC | Intergovernmental Panel on Climate Change |
| LC | Land Converted to Cropland |
| NDCs | Nationally Determined Contributions |

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
