# Peer review of "Assessing Carbon Emissions from Biomass Burning in Croplands in Burkina Faso, West Africa"

_fire, doi:10.3390/fire6100402_

Round 1
Reviewer 1 Report
Agricultural biomass burning is an important part in carbon emissions, leading it important to quantify its carbon emission especially in areas with little research. The manuscript literatured its experiment design and theory basis, covering two major climate zones with 96 experimental fires. Carbon emissions and emission factors of different land type in different climate zone are given, sheding light on scientific support for decision-making.
The reviewer appreciate the design and carry-out of the experiment in Bukina Faso. But the manuscript is far from reaching the standard of getting published. The English should be thoroughly checked and improved. All the expressions of equations should be re-written
Efforts were making for the preparation of the manuscript, but it appears that the authors were not systematically trained for writing research papers. Too many grammer errors, nonstandard expressions of equations, errors in figures have been found in the manuscript, hindering the reviewer from reviewing the manuscript and agreeing with the authors.
Specific comments
1. line 63: the English name for CO2 should be introduced at its first appearance, like in line 54.
2. Line 132 and from here on: the expression ‘cropland with <20 old years’ is nonstandard. ‘<’ can not be in a sentence. Rather, we use words to express this meaning.
3. Line 153: rewrite the sentence : ‘assessment (Figure3) . by adopting …’ This is grammerly wrong.
4. Figure 3: In the titles of the figure, there are (a) and (b), which cannot be found in the figure.
5. Line 221: ‘carbon content calculation ’ is the only content in Sect.2.7.2, so it’s not necessary to give it a section as 2.7.2.1.
6. Line 242: the sect.2.7.2.2 is missing
7. In Sect.2.7, the expressions of these equations are nonstandard. The authos should use Equation Editor tools, not type the letters in the manuscript. Things like ‘Ash% is not a scientific expression. Also ,we don’t use symbol ‘=’ when we explains the equation, like in line 215 and so on. What does ‘%Crem’ mean in equation (8)? What does the symbol ‘X’ mean in line 252? Does it mean the multiple sign? Then why is it different from the symbol ‘*’ in other equations.
8. Figure 5: The labels of sub-figures like (a)(b)(c) are at wrong positons, which should be at the left up corner of the sub-figure.
Too many grammer errors, nonstandard expressions of equations, errors in figures have been found in the manuscript, hindering the reviewer from reviewing the manuscript and agreeing with the authors.
Author Response
The authors would like to express their gratitude to the reviewers for their valuable comments and feedback. We acknowledge that all the comments contributed to substantially improving our manuscript. Here we respond to the questions and suggestions made by the reviewer, with a detailed description of the modifications made. All modifications are highlighted in red in the revised manuscript.
Comments and suggestions from reviewer
General comments and suggestions for authors
Comment 1: The reviewer appreciates the design and carry-out of the experiment in Burkina Faso. But the manuscript is far from reaching the standard of getting published. The English should be thoroughly checked and improved. All the expressions of equations should be re-written.
Efforts were making for the preparation of the manuscript, but it appears that the authors were not systematically trained for writing research papers. Too many grammar errors, nonstandard expressions of equations, errors in figures have been found in the manuscript, hindering the reviewer from reviewing the manuscript and agreeing with the authors.
Response: Thank you for these relevant comments. The English was thoroughly checked and improved by a native English-speaker. All the equations and expressions were also corrected and re-written in standard expressions as well as errors in figures (please see the corrections in the revised manuscript)
Specific comments from reviewer
Comment 1. line 63: the English name for CO2 should be introduced at its first appearance, like in line 54.
Response: We agree with this remark and have included the English name of Carbon dioxide (CO2) for its first appearance in the introduction.
Comment 2. Line 132 and from here on: the expression ‘cropland with <20 old years’ is nonstandard. ‘<’ cannot be in a sentence. Rather, we use words to express this meaning.
Response: Yes, we used words in the revised manuscript to explain the meaning of the expressions “Cropland with < 20 old years” and “Cropland with > 20 old years”. Specifically, “<”was replaced by “less than” and “>” by “more than”.
Comment 3. Line 153: rewrite the sentence: ‘assessment (Figure3). by adopting.. ’ This is grammar wrong.
Response: we removed the full stop mark “.” after “(Figure3)” and added “by” before adopting.
Comment 4. Figure 3: In the titles of the figure, there are (a) and (b), which cannot be found in the figure.
Response: (a) and (b) were added in the figure.
Comment 5. Line 221: ‘carbon content calculation’ is the only content in Sect.2.7.2, so it’s not necessary to give it a section as 2.7.2.1.
Response: Thank you for this remark. We deleted this section in the revised manuscript.
Comment 6. Line 242: the sect.2.7.3.2 is missing
Response: The sect 2.7.3.2 was added in the revised version.
Comment 7. In Sect.2.7, the expressions of these equations are nonstandard. The authors should use Equation Editor tools, not type the letters in the manuscript. Things like ‘Ash% is not a scientific expression. Also, we don’t use symbol ‘=’ when we explain the equation, like in line 215 and so on. What does ‘%Crem’ mean in equation (8)? What does the symbol ‘X’ mean in line 252? Does it mean the multiple sign? Then why is it different from the symbol ‘*’ in other equations.
Response: Thank you for these relevant comments. In the revised manuscript, we generated equations using Equation Editor tools as suggested, dealing with the confusion between symbol “X” and “*”. In addition, the expressions: Ash% and %Crem’ were replaced by “Ash (%)” and “Crem (%)” respectively. All similar expressions were corrected. The symbol “=” used to explain the abbreviations mentioned in equations were replaced by the symbol “:” throughout the revised manuscript.
Comment 8. Figure 5: The labels of sub-figures like (a) (b) (c) are at wrong positions, which should be at the left-up corner of the subfigure.
Response: All the labels of sub-figures are re-placed in the correct position as suggested.
Reviewer 2 Report
The authors presented an interesting paper. The authors present an important issue. , but:
I see a important problem with significant digits (significant figures) – for example:
19.16 ± 0.019
// it should be written 19.16 ± 0.02
or
894.575 ±0.599
There are many problems with significant digits in the paper.
The obtained CO2 emission results should be compared, for example, with annual CO2 emissions from cars (in Burkina Faso) to show the scale.
Author Response
The authors would like to express their gratitude to the reviewers for their valuable comments and feedback. We acknowledge that all the comments contributed to substantially improving our manuscript. Here we respond to the questions and suggestions made by the reviewer, with a detailed description of the modifications made. All modifications are highlighted in red in the revised manuscript.
Comments and suggestions from reviewer
Comment 1: The authors presented an interesting paper. The authors present an important issue, but: I see a important problem with significant digits (significant figures). For example: I see 19.16 ± 0.019. It should be written 19.16 ± 0.02 or 894.575 ±0.599.
Response: Thank you for the comment. In the revised manuscript, we made sure to round off the figures according to the standard rules.
Comment 2: The obtained CO2 emission results should be compared, for example, with annual CO2 emissions from cars (in Burkina Faso) to show the scale.
Response: Yes, it would be really interesting to compare our results with those of the transport sector. However, the existing estimates are based on default emission factors, making the comparison less realistic. In the future this comparison will be possible when specific emission factors per sector of activities are developed. According to existing data, CO2 emissions from transportation are estimated at around 2,039 kilotons in 2015.
Reviewer 3 Report
The manuscript studies a problem relevant to the circular economy and air quality. Determining emission factors from agricultural burning is important to quantify the problem and take mitigation actions.
Review the language throughout the document, avoiding redundancies.
It can be published due to its technical level, development, and discussion.
Congratulations.
Outstanding, little improvements are required.
Author Response
The authors would like to express their gratitude to the reviewers for their valuable comments and feedback. We acknowledge that all the comments contributed to substantially improving our manuscript. Here we respond to the questions and suggestions made by the reviewer, with a detailed description of the modifications made. All modifications are highlighted in red in the revised manuscript.
Comment: Review the language throughout the document, avoiding redundancies
Response: The manuscript has been revised by a native English speaker to improve the language. Thank you for your revision.